# Antarctic sea ice regime shift associated with decreasing zonal symmetry in the Southern Annular Mode

Serena Schroeter[1], Terence J. O'Kane[1,2], Paul A. Sandery[1]

[1]CSIRO Oceans and Atmosphere, Hobart, Tasmania, Australia
[2]Australian Centre for Excellence in Antarctic Science, Hobart, Tasmania, Australia

*Correspondence to*: Serena Schroeter (serena.schroeter@csiro.au)

**Abstract.** Across the long-term (~43 year) satellite record, Antarctic sea ice extent shows a small overall circumpolar increase, resulting from opposing regional sea ice concentration anomalies. Running short-term samples of the same sea ice concentration data, however, show that the long-term trend pattern is dominated by the earliest years of the satellite record.

Compensating regional anomalies diminish over time, and in the most recent decade, these tend towards spatial homogeneity instead. Running 30-year trends show the regional pattern of sea ice behaviour reversing over time; while in some regions, trend patterns abruptly shift in line with the record anomalous sea ice behaviour of recent years, in other regions a steady change predates these record anomalies. The shifting trend patterns in many regions are co-located with enhanced north-south flow due to an increasingly wave-3-like structure of the Southern Annular Mode. Sea surface temperature anomalies also shift

from a circumpolar cooling to a regional pattern that resembles the increasingly asymmetric structure of the Southern Annular Mode, with warming in regions of previously increasing sea ice such as the Ross Sea.

## 1 Introduction

Contrary to expectations of sea ice coverage under a warming climate, sea ice in the Southern Ocean has not decreased over the satellite record as it has in the Arctic Ocean; rather, large diametrically opposed regional sea ice trends have resulted in a

small overall increase in Antarctic sea ice extent (Turner et al., 2015; Simmonds, 2015; Parkinson, 2019; Comiso et al., 2017). A small, steady increase in areal coverage between 1979-1999 was followed by an accelerated increase between 2000-2014 (Meehl et al., 2019), culminating in record high sea ice extent in 2014 (Comiso et al., 2017) before an unprecedentedly steep spring retreat in 2016 (Turner et al., 2017), and record low ice between 2017 and 2019 (Parkinson, 2019; Raphael and Handcock, 2022). While Antarctic sea ice returned to near average coverage in 2020, sea ice extent during the summer of

2021-22 reached the lowest point of the ~43-year satellite record (Raphael and Handcock, 2022; Wang et al., 2022; Turner et al., 2022). Sea ice is sensitive to changes in and interactions between the atmosphere and ocean (Hobbs et al., 2016), and indeed both atmospheric and oceanic influences have been identified to help explain the recent erratic behaviour (Turner et al., 2017; Stuecker et al., 2017; Wang et al., 2019; Schlosser et al., 2018; Meehl et al., 2019; Wang et al., 2022; Schossler et al., 2020). However, consensus is still lacking for not only the processes and mechanisms driving historical sea ice trends, but

also the dominant drivers of internal variability (Eayrs et al., 2019; Eayrs et al., 2021). Low confidence is currently placed in the simulation of mean and historical Antarctic sea ice in global coupled climate models, which tend not to accurately capture

the spatial heterogeneity or the seasonality of observed sea ice trends (Turner et al., 2013a; Hobbs et al., 2016; Eayrs et al., 2019). The disparity between simulated and observed sea ice is further complicated by the non-linearity of sea ice trends on both short-term and decadal scales (Handcock and Raphael, 2020; Eayrs et al., 2021). Additionally, the extent to which internal variability can account for the observed increase (as well as the disparity between observed and simulated trends) remains contested (Turner et al., 2013a; Mahlstein et al., 2013; Shu et al., 2015; Shu et al., 2020; Roach et al., 2020).

Recent examinations of Antarctic sea ice cover have generally considered the mean state and trends over the long-term, continuous satellite record (e.g. Eayrs et al., 2021; Raphael and Handcock, 2022; Parkinson, 2019). Climatological averaging smooths internal variability (which for Antarctic sea ice is known to be high), and is taken as reflective of the expected background state of Antarctic sea ice from which anomalies can be calculated. In a stationary or slowly-evolving variable, the climatological estimate can quite closely align with datapoints throughout the averaging period (Livezey et al., 2007). However, if significant trends exist in the averaging period, the background estimate may not be a realistic expectation from which future projections can be calculated (Arguez and Vose, 2011). It has long been noted that significant trends have existed during periods of the satellite record (Parkinson, 2019; Handcock and Raphael, 2020; Eayrs et al., 2021), which potentially shift the background estimate over time. Sea ice is strongly affected by atmospheric changes such as temperature, for which shorter 15-year temporal averages for climatological norms have been suggested (Arguez and Vose, 2011). Furthermore, while much of the literature reviewing Antarctic sea ice condenses metrics such as extent or areal cover into spatial averages or sums, either as a circumpolar total or a sector-based approach utilising geographical constraints (e.g. Zwally et al., 1983; Parkinson and Cavalieri, 2012; Parkinson, 2019) or spatial-autocorrelation dividing sea ice into regions of distinct interannual variability (e.g. Raphael and Hobbs, 2014), Antarctic sea ice trends are spatially heterogeneous, and large-scale averaging may neglect important small-scale regional trend characteristics.

This study presents a detailed examination of the temporal and spatial changes in Antarctic sea ice variability, comparing 15-year running samples of sea ice concentration with samples of large-scale atmospheric and oceanic variability patterns. Data and methods are described in Section 2, with results illustrated in Section 3. Discussion and concluding remarks are presented in Section 4.

## 2 Method

The monthly mean passive microwave satellite sea ice concentration data from the Climate Data Record are obtained from the National Snow and Ice Data Center (NSIDC), encompassing January 1979 – May 2021 (Meier et al., 2021b; Peng et al., 2013). This timeseries is then supplemented with monthly averages of the daily near-real-time Climate Data Record (Meier et al., 2021a) for June – December 2021. Satellite data is obtained on the native equal-area 25km polar stereographic grid (https://nsidc.org/data/ease). In order to obtain estimates of sea ice area and extent at each longitude point, the sea ice concentration (SIC) data are reprojected and regridded onto a regular 1° latitude x 1° longitude grid. Following convention, at each longitude, sea ice extent (SIE) is calculated as the sum of the product of grid cell area for cells with at least 15% ice

coverage, and sea ice area (SIA) is calculated as the sum of the product of grid cell area and ice area fraction (e.g. Roach et al., 2020).

Monthly averaged mean sea level pressure (SLP) between January 1979 – December 2021 from the Japanese Meteorological Agency (JMA) global atmospheric reanalysis JRA-55 (Kobayashi et al. 2015) are obtained from the Research Data Archive at the National Center for Atmospheric Research, Computational and Information Systems Laboratory (https://doi.org/10.5065/D60G3H5B). Sea surface temperature (SST) data is obtained from the Met Office Hadley Centre (https://www.metoffice.gov.uk/hadobs/hadisst/) Sea Ice and Sea Surface Temperature dataset version 2 (HadISST.2.2.0.0, hereafter HadISST2 (Titchner and Rayner, 2014)). HadISST2 is provided by the Met Office on a 1° latitude by 1° longitude grid; the 1.25° JRA-55 data are regridded onto the same grid for consistency in comparing with SIC and SST.

To prepare data for analysis, each dataset is sliced into running 15-year (e.g. January 1979-December 1993, January 1980-December 1994 etc.) or 30-year samples depending on the context, from which climatological means and monthly anomalies are calculated. Eigenvectors are produced via empirical orthogonal analysis (EOF) to show patterns of maximum variance in SIC, SIA and SLP; these EOF patterns are calculated from monthly and seasonal anomalies from means of the 15-year samples (following detrending and weighting by cosine of latitude, to compensate for the convergence of meridians near the pole). Trends are calculated using linear least-squares regression, and the resulting residual variance and standard error calculations modified to account for temporal autocorrelation using an effective sample size based on lag-1 autocorrelation coefficients (Santer et al. 2000). Samples and regions where the trend is statistically significant are shown by overlaid stippling.

## 3 Results

### 3.1 Variability and trends in Antarctic sea ice

The large annual cycle of Antarctic sea ice (Figure 1a) has shifted over time, with running 15-year monthly differences of average sea ice extent (SIE) and area (SIA) from the 1979-2021 mean clearly showing a steady increase over the satellite record that reverses in the most recent samples (Figure 1b-c). Given the anomalously low sea ice occurring between 2016-2021 (described in Section 1), these shorter-than-typical climatologies merely highlight the nullifying impact of recent record low sea ice events. However, the proportion of 15-year samples exceeding the 1979-2021 mean is greater than half year-round, indicating that despite recent record low sea ice, the long-term mean of both SIA and SIE is dominated by the earliest years. The disproportionality of samples above the mean is particularly high during late summer and autumn (e.g. 22 of 29 SIA samples above the 1979-2021 mean in April-May and December), which is also when the largest sea ice trends have been observed (Hobbs et al., 2016; Holland, 2014; Eayrs et al., 2019). Calculating anomalies from a biased long-term sea ice average may obscure or misrepresent variability at short-term timescales, hindering robust interpretation of sea ice behaviour and trends. We posit that the magnitude or even sign of observed anomalous sea ice behaviour changes substantially under different timespans: for example, if sea ice anomalies were calculated from a sample encompassing only the most recent decades, the

short-term climatologies imply that the negative anomalies since 2016 would appear even more extreme, as mean sea ice is of a higher magnitude than the mean of the ~43 year climatology.

Spatially, the increasing mean SIE and SIA described above is most notable from west of the Dumont D'Urville Sea to the Ross Sea, as well as in the Haakon VII Sea, while mean sea ice steadily decreases in the Bellingshausen and western Weddell seas (Figure 1d-e). Compensating anomalies and opposing trends have long been noted in the aforementioned regions (Parkinson, 2019; Parkinson and Cavalieri, 2012; Comiso et al., 2017; Turner et al., 2015; Simmonds, 2015), with a 'cancellation' effect of negative regional anomalies reducing the magnitude of the positive regional anomalies to produce a smaller circumpolar value. Despite statistically significant positive trends in Antarctic sea ice existing at some points during the satellite record, strong non-linearity occurs within the trend due to high decadal and sub-decadal variability (Handcock and Raphael, 2020). For example, trends of SIE and SIA between 1979 and 2021 are positive (but statistically insignificant): 7.3 x $10^3$ km$^2$ per year for SIE (p>0.17) and 5.7 x $10^3$ km$^2$ per year for SIA (p>0.2) (Figure 2a); trends of annual circumpolar SIE and SIA anomalies for only the most recent 30 years (1992-2021) are negative (also statistically insignificant): -4.3 x $10^3$ km$^2$ per year for SIE and -3.5 x $10^3$ km$^2$ for SIA (both p>0.6).

Regional variability (shown by the standard deviation of annual SIE and SIA anomalies over the longitudinal domain) shows peaks mostly in the earlier half of the satellite record (e.g. 1980, 1985-6, 1992, 1999, 2003) interspersed with troughs (1984, 1993, 1997) indicating higher year-to-year variation in opposing regional anomalies (Figure 2b). Since 2004, however, regional variability is comparatively moderate with much smaller year-to-year variation, and a small decline between 2018-2021 implies a relatively persistent reduction in opposing anomalies. Contours of annual SIA anomalies by region (Figure 2c; SIA is presented instead of SIE as it is less noisy, though both show the same decreasing variability pattern) indeed show that strongly opposing regional anomalies tended to occur earlier in the satellite record. In much of the most recent decade, however, anomalies have veered toward the same sign in the Haakon VII, Ross and Weddell regions (Figure 2c), so despite the anomalies tending to be lower magnitude than in the 'peaks' of earlier years, the compounding effect of spatially-homogeneous anomalies rather than spatially-compensating anomalies leads to higher-magnitude annual average anomalies overall (Figure 2a).

Eigenvectors produced through empirical orthogonal analysis (EOFs; also known as principal component analysis) show the spatial pattern that accounts for the maximum proportion of variance in detrended anomalies. The first (highest-variance-explained) EOF pattern for the earliest sample (all monthly anomalies between 1979-1993) of sea ice concentration (SIC) in all four seasons shows opposing anomalies between the Ross/Amundsen and Weddell seas (Figure 3 a-d), while JJA and SON also have opposing anomalies around the rest of the continent in a wave-3-like pattern (Figure 3c, d). The most recent sample (2007-2021) produces a very different pattern during DJF and MAM, in which the opposing anomalies weaken and diminish spatially, and anomalies of the same sign appear around much of the continent instead (Figure 3e, f), with trends calculated across the 15-year samples tending towards increasing zonal symmetry (which is statistically significant (p<0.05) in the Ross Sea region during MAM) (Figure 3i, j). During JJA and SON, the opposing anomalies in the earliest patterns of maximum variance (Figure 3c, d) persist (though more weakly) in the more recent samples (Figure 3g, h). The weakening of the dipole

pattern of anomalies between the Ross/Amundsen and Weddell seas during JJA is also statistically significant in part of the

Ross Sea and around the Antarctic Peninsula (Figure 3k). As the sign of anomalies is fairly consistent at each longitude, we
show the steady reduction of the pattern of opposing anomalies over time by averaging EOF1 across the latitudes at each
longitude for all monthly anomalies in each 15-year sample (Figure 3m). The results emphasise growing spatial homogeny in
the pattern of anomalies around much of East Antarctica, a steady diminution through time of opposing anomalies between
the Ross/Amundsen seas and Bellinghausen Sea, and an eastward shift of the location of maximum 'opposing' anomalies

from a dipole between the Ross/Amundsen and Weddell seas, to the Amundsen/Bellingshausen and Weddell seas. The
changing pattern of EOF1 of SIC, particularly the eastward shift of the dipole pattern, alludes to potential changes in regional
trend patterns that may be visible on shorter timescales. The high internal variability of sea ice, however, precludes the use of
15-year samples of sea ice anomalies to calculate discernible trends, and we utilise 30-year samples instead to calculate running
trend patterns.

Trends of anomalies from 30-year samples reveals a very different picture than the long-term trend implies (Figure 4). Annual
average trends show that the compensating dipole of regional trends described above (positive in the Ross, central and eastern
Weddell and western Haakon VII seas, negative in the Davis, Amundsen and Bellingshausen seas) is substantially dampened
over time (Figure 4a). Diminishing negative trends in the Amundsen, Bellingshausen and western Weddell Sea are consistent
through time, clearly pre-dating the recent anomalously low-sea ice years. In contrast, changing patterns in the Ross, eastern

Weddell and Haakon VII seas are noisy in the earliest samples but abruptly shift in samples that coincide with the anomalously
sharp sea ice retreat in 2016 and beyond, which has been attributed to unusual atmospheric and oceanic conditions in the
Southern Ocean (Turner et al., 2017; Stuecker et al., 2017; Wang et al., 2019; Schlosser et al., 2018; Meehl et al., 2019).
From a seasonal perspective, trend patterns clearly differ between DJF/MAM and JJA/SON, with the regionally opposing
trends much stronger in the former (Figure 4b-e). Statistically significant ($p<0.05$) increasing trends in the western and central

Ross Sea occur throughout the first two-thirds of the samples in DJF and MAM, but disappear in the most recent samples
(Figure 4b-c). A persistent decreasing trend in the eastern Ross/western Amundsen seas intensifies in recent samples in DJF,
becoming statistically significant in the two most recent 30-year periods, while an increasing trend in the eastern
Amundsen/western Bellingshausen seas appears in DJF in samples from 1987-2016 onwards (also statistically significant in
the two most recent samples). The Weddell Sea has a positive trend throughout DJF and MAM samples (statistically significant

along the western boundary in the more recent half of the DJF samples), while the sign of the trend in the central Weddell Sea
switches to negative (statistically insignificant) from 1990-2019 onwards. By comparison, while the magnitude of the trends
in JJA and SON is smaller, an eastward shift is evident from positive, statistically significant trends in the western and central
Ross Sea in the first half of the samples towards positive (but statistically insignificant) trends in the Amundsen/eastern
Bellingshausen seas in the second half of the samples (Figure 4d-e). Instead, in the 5 most recent patterns, the western/central

Ross Sea has weakly negative or neutral trends in JJA and SON. Those same 5 JJA and SON samples also shift towards more
wave-3-like trend patterns, positive over the HaakonVII, Dumont D'Urville and Amundsen/Bellingshausen seas and negative
in the Davis, Ross and Weddell seas. Statistically insignificant negative winter and spring trends in the Weddell Sea are fairly

consistent across the samples, contrasting with large (statistically significant at times) positive trends in DJF and MAM for most samples. Indeed, it is only in the most recent 3 samples that the DJF and MAM trends turn negative (statistically insignificant), indicating the effect of the record low ice of recent years and, with a persistent, year-round negative trend, explains the sudden high-magnitude negative annual trend patterns for those samples, reaching almost the same strength as the earliest negative annual trend in the Bellingshausen Sea (Figure 4a).

The eastward shift of positive sea ice anomalies from the Ross Sea to the Amundsen/Bellingshausen seas could be due to several potential drivers, such as the strength and latitudinal/longitudinal position of the climatological low-pressure centre in the Amundsen Sea (the Amundsen Sea Low (Fogt et al., 2012b; Raphael et al., 2015; Turner et al., 2013b)), which impacts seasonal and interannual sea ice variability (Raphael et al., 2019), low-frequency large-scale atmospheric forcing and high-frequency synoptic variability (weather) (Matear et al., 2015), and decadal-scale ocean drivers such as anomalous subsurface ocean cooling via the Antarctic Circumpolar Current (Morioka et al., 2022). It has long been understood that the complex temporal sea ice variability on seasonal, multi-year, decadal and multi-decadal scales as well as the spatially heterogeneous relationships between Antarctic sea ice in various sectors with different atmospheric and oceanic drivers indicates Antarctic sea ice variability and trends are unlikely to be associated with a single driver (Holland, 2014). Recently, however, Kimura et al. (2022) found similar contrasting annual trends between 2003-2019 of increasing SIC in the Amundsen and Bellingshausen seas and parts of East Antarctica and decreasing SIC elsewhere, and demonstrated the dominance of dynamic wind forcing in driving seasonal evolution of sea ice in the regions of increasing SIC. Seasonal and regional variability and trends in Antarctic sea ice are strongly influenced by wind, through both thermodynamic and dynamic effects (e.g. Hobbs et al., 2016; Raphael and Hobbs, 2014; Schroeter et al., 2017; Matear et al., 2015; Holland and Kwok, 2012; Haumann et al., 2014; Hall and Visbeck, 2002; Bernades Pezza et al., 2012; Stammerjohn et al., 2008). We turn now to short-term variability and trends in the dominant mode of Southern Hemisphere atmospheric variability, the Southern Annular Mode (SAM).

### 3.2 Meridional structure shift in the Southern Annular Mode

Atmospheric circulation in the Southern Hemisphere high latitudes is predominantly zonally-symmetric, with a strong westerly polar jet encircling the continent near 60°S (Kidson and Sinclair, 1995). SAM is often described as a predominantly zonally-symmetric, annular (ring-like) structure of opposing anomalies between the mid- and high-latitudes (Gong and Wang, 1999; Thompson and Wallace, 2000; Marshall, 2003). However, there are mechanisms of asymmetric flow embedded within the otherwise zonal atmospheric circulation (Van Loon, 1972), such as poleward propagating Rossby wave trains from the tropics known as the Pacific South-American (PSA) modes (Mo and Higgins, 1998; Mo and Paegle, 2001), and the quasi-stationary zonal waves 1 and 3 (ZW3) (Hobbs and Raphael, 2010; Raphael, 2004). A zonal wave-3 pattern is embedded with the SAM itself, with the annular pattern around Antarctica containing three non-annular components, juxtaposed with three non-annular components of opposing anomalies in the mid-latitude belt (Figure 5a). SAM has both a positive (deepening pressure anomalies over the high-latitudes and increasingly high pressure anomalies over the mid-latitudes) and a negative (vice versa) phase polarity, and trends calculated since 1979 show SAM tending towards its positive phase (Fogt and Marshall, 2020). During

the positive SAM phase, negative pressure anomalies occur over the Antarctic region, projecting onto the ASL, resulting in enhanced local meridional winds and disparate regional sea ice behaviour in the Ross, Amundsen, Bellingshausen and Weddell seas (Holland et al., 2017). Positive SAM is also associated with a strengthening of the westerly polar jet, which is evident in 15-year running samples of 850hPa zonal winds (see Figure S1). An increasingly annular pattern of atmospheric flow implies stronger zonal winds over the sea ice zone (outside the ASL region), and would tend to produce more northward sea ice transport through the Ekman effect (Hall and Visbeck, 2002). However, in recent years, the structure of the SAM has become increasingly asymmetric (Wachter et al., 2020; Campitelli et al., 2022), indicating more north-south airflow around the continent and, as a result, more spatially heterogeneous sea ice behaviour. In EOF1 of monthly detrended SLP anomalies between 1979 and 1993 (Figure 5b), the ring-like, annular structure of SAM is evident, moreso than in the long-term mean (Figure 5a). In more recent 15-year samples (e.g. 2007-2021; Figure 5c), though, the annular structure becomes more wave-like, as the non-annular components in the mid-latitudes contract southwards. The difference between early samples and more recent samples (Figure 5d) essentially reproduces the pattern of the asymmetric SAM component at 700hPa (Campitelli et al., 2022). The asymmetric SAM component is calculated as anomalies from the zonal mean of the SAM pattern, and a spatial average calculated for each longitude between -70°S and -55°S to encompass the region of maximum ZW3 amplitude and much of the sea ice zone (Campitelli et al., 2022). The resulting longitudinal averages shows a consistent shift in structure from annular to increasingly wave-3-like (Figure 5e). The deepening and longitudinal contraction of the non-annular component in the Amundsen Sea positions it further eastward, away from the Ross Sea, with a clear reduction in cyclonic flow over the west of this region (Figure 5c).

The spatial pattern of SAM is more zonally-symmetric when calculated from monthly or annual mean anomalies rather than seasonal mean anomalies, and it is important to note that the structure of SAM has substantial seasonal and decadal variability (Fogt et al., 2012a; Fogt and Marshall, 2020). Seasonal EOFs spanning the long-term record (1979-2021) demonstrate the strongest zonally-symmetric flow during the summer months (Figure 6a), and greater asymmetric flow during the austral autumn, winter and spring (Figure 6e, i, m). Trends calculated across running EOFs of 15-year samples of detrended DJF anomalies (Figure 6d) show a reduction of the non-annular components seen in earlier samples (Figure 6b) towards an intensification of the zonally-symmetric pattern in the high- and mid-latitude bands in the most recent patterns (Figure 6c), in line with trends identified in other studies (Fogt et al., 2012a; Fogt and Marshall, 2020). Though the aforementioned studies also found a smaller trend towards a more annular spatial pattern of SAM during the austral autumn, the trend pattern across the 15-year MAM sample patterns (Figure 6h) shows that, at least on shorter timescales, there is a diminution of the strength of the annular pattern of opposing anomalies in the mid- and high-latitude belts in early samples (Figure 6e), tending towards a strong dipole in an otherwise relatively weak asymmetric pattern of anomalies in the sample encompassing 2007-2021 (Figure 6g). The strength of the asymmetric intensification of winter (JJA) and spring (SON) spatial patterns of SAM in winter is substantially stronger (Figure 6j, k, n, o), with trends of the cool season 15-year EOFs demonstrating a clear wave-3-like pattern of strengthening non-annular components in both the mid- and high-latitude belts (Figure 6l, p).

Sea ice is most sensitive to the influence of SAM (particularly meridional forcing from its non-annular components and projection upon the ASL) during the seasons of JJA and SON, particularly the Ross and Bellingshausen seas (Simpkins et al., 2012). As shown in Section 3.1, the sign of sea ice trends in the Ross, Amundsen, Bellingshausen and western Weddell seas has essentially reversed throughout the samples, which in the Amundsen and Bellingshausen is a gradual progression that precedes the recent record anomalous sea ice behaviour (Figure 4a). The slope of the 30-year SIC trend patterns (the trend of the trends) is overlaid with contours showing the difference between the earliest and most recent pattern of the asymmetric SAM component (which Figure 5e shows to be fairly consistent and gradual around much of the continent) for the average seasons of December to May (DJFMAM) and June to November (JJASON) (as in Arblaster and Meehl (2006)) (Figure 7). Negative values (shown by dashed contour lines) indicate increasing anticyclonic flow, and positive values the reverse. In both seasons, several regions show striking consistency between the slope of SIC trends and changes in the asymmetric SAM flow pattern. In the warmer months (DJFMAM), the negative slope of trend patterns in the Ross Sea is co-located with an increasingly anti-cyclonic pattern of asymmetric SAM flow (Figure 7a). At the western edge of the Ross Sea, strengthened northerly flow suggests warmer surface temperatures aiding summer melt and dynamic effects of constraining equatorward ice transport; at the eastern boundary between the Ross and Amundsen Seas, the enhanced southerly flow coincides with a negative SIC trend slope at the coast and positive nearer the ice edge, implying a dynamic effect of pushing more ice northward. Across the Amundsen and Bellingshausen, the cyclonic flow of asymmetric SAM is strengthened and shifted eastward, co-located with a positive SIC trend slope. Easterly flow over the Bellingshausen implies coastward constraining of the ice, while the thermodynamic and dynamic effects of northerly flow from the juxtaposition of the edges of the cyclonic and anticyclonic flow patterns are shifted eastward towards the Weddell Sea, where ice is compacted along the western boundary but reduced in the centre and eastwards over the Haakon VII Sea.

During the cool season months, JJASON, the intensified SAM asymmetry is similarly co-located with changes in SIC trends around much of the continent (Figure 7b). Increasing SIC around the Dumont D'Urville region is overlaid by increasing northerly flow; Kimura et al. (2022) suggest this region is sensitive to dynamic changes more than thermodynamic changes, and the pattern implies that the increasing SIC slope in this part of East Antarctica could result from coastward compaction of ice. A southward and longitudinal contraction of the anticyclonic flow over the Ross Sea and the juxtaposition of the opposing flow patterns of the Ross and Amundsen seas suggests enhanced northward flow and associated equatorward ice transport in the eastern Ross/western Amundsen, coincident with a positive SIC slope. By comparison, the eastward shift of this pattern and enhanced northerly flow over the western Ross Sea is co-located with a negative slope of SIC trends.

Not all regions around Antarctica show a clear relationship with the changes in asymmetric SAM, however. For example, during JJASON, strengthening mid-latitude anomalies north of the Weddell Sea result in enhanced westerly flow across the north of the Weddell region and diminished northerly flow at the western edge (Figure 7b). This enables SIC to increase north of the western boundary; however, SIC decreases in the central and eastern Weddell Sea. Decadal sea ice variability is known to be high in this region (e.g. see annual average SIA anomalies in Figure 2c). Multiple mechanisms have been identified as drivers of Weddell Sea decadal sea ice variability, including the strength of westerly winds: over the satellite period, sea ice

increases in the Weddell Sea have been preceded by weakened zonal wind forcing towards the north, which leads to a reduction in equatorward ocean currents and less Ekman upwelling of warm water, producing cooler temperatures in the upper ocean and conditions favourable for sea ice production (Morioka and Behera, 2021). The increasing zonal flow the JJASON asymmetric SAM pattern (Figure 7b) suggests the opposite of this pattern, though, which implies increased northward currents, enhanced upwelling and warmer upper ocean temperature, with subsequent decreases in ice coverage. However, on longer timescales (~20 years), sea ice in the Weddell Sea is also influenced by local atmosphere-ice-ocean interactions, in which decreased sea ice coverage in the Weddell Sea permits greater surface evaporation and thus higher salinity in the upper ocean, strengthening the Weddell Gyre which in turn reduces upper ocean temperatures, producing conditions favourable for sea ice expansion (Morioka and Behera, 2021). Multiple analyses have shown that sea ice trends in the Weddell Sea are influenced by a combination of large-scale atmospheric forcing (including SAM) and synoptic-scale variability (Matear et al., 2015; Schroeter et al., 2017; Raphael and Hobbs, 2014), so in this region at least, the effect of a shifting SAM structure may be more tempered by other drivers than in other areas of West Antarctica upon which SAM has a strong influence. Similarly, around much of East Antarctica, sea ice anomalies are known to be dominated by synoptic variability and cyclonic activity rather than large-scale atmospheric modes (Matear et al., 2015; Schroeter et al., 2017; Turner et al., 2015) and therefore, despite increasing SAM asymmetry in these regions, sea ice trends between the Haakon VII and Davis seas show no clear corresponding response (Figure 7b).

### 3.3 Changes in Southern Ocean Sea Surface Temperature

Climatological reconstructions suggest that sea surface temperature (SST) across the Southern Ocean increased across much of the early- and mid-20th century before exhibiting a reversal in the late 1970s (Turney et al., 2017). In the decades following, a regime shift occurs, towards a "warm state" in which the Indian Ocean and western boundary currents of the Southern Ocean warm but the high-latitude SSTs cool (Freitas et al., 2015). In line with the increase in mean Antarctic sea ice since 1979, high-latitude SSTs in the Southern Ocean have decreased, associated with strengthening westerly winds and negative pressure anomalies (Fan et al., 2014; Armour et al., 2016; Blanchard-Wrigglesworth et al., 2021). The shift to positive SAM is also associated with thermodynamic atmospheric changes, such as enhanced cooling and drying of the atmosphere overlying the high-latitude Southern Ocean, and reduced downwelling longwave radiation, which lead to negative SST anomalies (Kusahara et al., 2017; Doddridge and Marshall, 2017). A nonmonotonic, two-timescale Southern Ocean SST response to SAM is seen in many global coupled climate models, in which abrupt SST cooling through anomalous equatorward Ekman drift occurs under a transition to positive SAM, but gradually reverts towards a warming signal due to either advective or mixing-based mechanisms over the longer-term (Kostov et al., 2017; Ferreira et al., 2015; Seviour et al., 2019; Doddridge et al., 2021). However, a warming SST signal has eluded the observational record thus far, with mechanisms such as eddy compensation potentially dampening the effect of wind-driven upwelling (Doddridge et al., 2019). Nonetheless, recent work shows that observed sea ice trend patterns cannot be explained by winds alone, requiring nudging of SST north of the sea ice edge towards observations in order to reproduce observed trends (Blanchard-Wrigglesworth et al., 2021). Given the close association of SST

in the region north of the sea ice edge to both sea ice anomalies and SAM, and the shifting patterns of both, we now examine running anomalies of SST.

Although the long-term (1979-2021) trend of sea surface temperature (SST) within the sea ice zone and just north of the sea ice edge is negative, mid-latitude SSTs are increasing (Figure 8a). Considering the regression coefficients of only the most recent 30-year period (1992-2021), the trend pattern is less widespread around the western Ross Sea, but more so in the Amundsen and Bellingshausen seas (Figure 8b). Over the last 15 years (2007-2021), however, a wave-3-like pattern of SST anomalies appears instead (Figure 8c), bearing resemblance to the enhanced meridional pattern of SAM (Figure 5d).

Regression coefficients of 30-year samples of SST anomalies north of the ice edge (-55°S to -40°S) show a reversal of the dipole between the region above the western and eastern Ross Sea; whereas in early 30-year samples the region north of the western Ross Sea was clearly cooling compared to warming to the east, SSTs are increasing across the region in the most recent sample (Figure 8d). SSTs are also increasing north of the Bellingshausen Sea, but declining sharply in recent samples across the neighbouring region above the Amundsen Sea. North of the Weddell Sea, 30-year trends of SSTs above the sea ice zone are decreasing to the west and showing little clear tendency above the central Weddell Sea. Only in the far east, adjacent to Haakon VII Sea, do SST trends show an increasing tendency before dropping again in line with enhanced southerly flow under the increased asymmetry of SAM.

## 4 Discussion and Conclusions

We use short-term running means and anomalies to show that, while climatological Antarctic sea ice has increased over the satellite record, the long-term pattern is dominated by the earliest years, and is reversing across much of the sea ice zone, particularly in the Haakon VII, Ross and Bellingshausen seas. The long-term trend of circumpolar sea ice anomalies shows a small expansion over time, which has long been cited as the result of high-magnitude opposing regional anomalies; however, the regional sea ice dipole decreases over the past decade, with smaller but more spatially homogeneous compounding anomalies. Trends of anomalies based on running 30-year samples reveal a reversal of both the increasing trend in the Ross, Weddell and Haakon VII seas and the decreasing trend in the Amundsen and Bellingshausen seas. Indeed, the most recent 30-year sample shows negative trends in the Ross and eastern Weddell seas, positive trends in the Amundsen and western Weddell seas, and close to zero trend in the Bellingshausen Sea. Though some of these trend patterns shift more abruptly due to the recent anomalous sea ice declines, the reversal in other regions has been consistent across the samples.

It is clear that the long-term circumpolar sea ice increase has been abruptly interrupted by the record low sea ice between 2016-2019, with the rate of decline over those few years equalling the rate of decline over 30 years in the Arctic (Handcock and Raphael, 2020; Parkinson, 2019; Eayrs et al., 2021). However, several studies using ice cores and whaling records also identified sharp sea ice declines in the decades preceding the satellite sea ice record, particularly in the late 1970s just prior to the launch of the satellite record (Kukla and Gavin, 1981; De La Mare, 1997; Curran et al., 2003; De La Mare, 2009; Cotte and Guinet, 2007). A recent reconstruction suggests that Antarctic sea ice declined through much of the period from 1905 until

just prior to satellite observations, particularly in the Ross, Amundsen and Haakon VII seas, and that the observed overall increasing trend since 1979 is unique in the context of the twentieth century (Fogt et al., 2022). Simulations investigating multi-decadal sea ice trends which reproduced the small overall increase in Antarctic sea ice also produced decreasing sea ice in the decades prior to the satellite record (Goosse et al., 2009). That these sea ice declines were then followed by the observed overall increase for several decades raises speculation as to whether Antarctic sea ice is simply very responsive to sharp changes in atmospheric or oceanic conditions, and is capable of recovering from those declines over the ensuing decades (Parkinson, 2019; Eayrs et al., 2021).

As the dominant mode of large-scale atmospheric circulation in the Southern Hemisphere, SAM has a strong impact on sea ice, particularly in the West Antarctic region (Lefebvre and Goosse, 2005; Lefebvre et al., 2004; Holland et al., 2017; Doddridge and Marshall, 2017; Stammerjohn et al., 2008; Schroeter et al., 2017; Raphael and Hobbs, 2014; O'kane et al., 2013b). Several studies have noted that, prior to 1978, Southern Hemisphere mid-latitude circulation was dominated by a metastable blocking regime, in which zonally symmetric circulation is suppressed in favour of heightened meridional motion (O'kane et al., 2013a; Freitas et al., 2015). Following the late 1970s, however, a regime shift occurred in all seasons in which SAM (which was previously associated with a transition state) replaced the negative phase of this blocking pattern, increasing zonally-symmetric (annular) circulation in both frequency and intensity and reducing the occurrence and persistence of blocking (O'kane et al., 2013a; Freitas et al., 2015; Franzke et al., 2015). Surface air temperatures and Southern Ocean SSTs reflected the weakened blocking state (Freitas et al., 2015; O'kane et al., 2013b; Franzke et al., 2015), and the strength of the subtropical jet decreased while upper tropospheric zonal wind strength increased north of the sea ice zone (Frederiksen et al., 2011; Frederiksen and Frederiksen, 2007).

The shift towards positive SAM polarity in the austral summer months is often described as predominantly a forced response to external forcing from ozone depletion in the stratosphere and upper troposphere, and the combination of ozone depletion and increasing greenhouse gases in the lower troposphere, with smaller contributions of varying sign from natural sources (such as solar forcing and volcanic aerosols) and anthropogenic sulfate aerosols (Arblaster and Meehl, 2006; Fogt and Marshall, 2020). However, initial signs of recovery have been detected in ozone levels in the Antarctic stratosphere (Solomon et al., 2016), and as ozone levels recover, a previously complementary relationship between ozone depletion and increasing greenhouse gases driving zonally-symmetric atmospheric forcing in the mid- and high-latitudes of the Southern Hemisphere will presumably become compensatory instead (Polvani et al., 2011). Model studies project that, depending on the level of emissions mitigation, continuing greenhouse gas increases and the associated radiative forcing can maintain the shift towards positive SAM (Shindell and Schmidt, 2004; Arblaster and Meehl, 2006; Holland et al., 2022). Furthermore, the atmospheric response of greenhouse gas and solar forcing contains more asymmetry than that of ozone (Arblaster and Meehl, 2006), and the effect of ozone is limited largely to the warmer months of the year, whereas greenhouse gases dominate when all seasons are considered (Franzke et al., 2015).

As SAM has a zonal wave-3 structure embedded within it (Van Loon, 1972), the intensifying annular mode has in recent years undergone a shift towards its asymmetric component (Wachter et al., 2020; Campitelli et al., 2022), increasing meridional

(north/south) flow over the sea ice zone. We show the co-location of spatial changes in the asymmetric component of SAM (and the corresponding north-south wind changes) with regions from Dumont D'Urville Sea to the Antarctic Peninsula where much of the spatial pattern of sea ice trends is reversing. Annual regional anomalies present a somewhat counterintuitive

narrative, changing over time from strongly opposing anomalies (producing small circumpolar average anomalies) to more spatially homogeneous sea ice anomalies (which compound to large averages), despite the increasingly wave-3-like pattern of SAM. However, the pattern of annual anomalies does hint to the changing spatial asymmetry of SAM and SSTs, with consistent sign of anomalies in the eastern Haakon VII, Ross and Weddell seas where negative pressure is intensifying, and relatively small or neutral anomalies in the intervening regions where the wave-3 pattern is tending towards positive as the mid-latitude

pressure anomalies shift south. Seasonal trends show a clear, opposing wave-3 trend pattern emerging during winter and spring, when the link between SAM and sea ice is strongest (Simpkins et al., 2012), which is diminished through annual averaging by opposing trends during summer and autumn. We also show that the increasing asymmetry of SAM does not fit well to sea ice anomalies around parts of East Antarctica and the Weddell Sea that are known to be dominated by other mechanisms (such as weather, tropical forcing etc), indicating that the influence of a changing SAM structure is most important in the regions (such

as the Ross, Amundsen and Bellingshausen seas) where SAM is known to strongly influence sea ice variability.

Though rebounding of sea ice has followed sharp declines in the past, an increase in the occurrence and intensity of the positive phase of SAM leads to increased temperatures in the subsurface ocean, as well as increased upwelling close to the continent (Verfaillie et al., 2022), with substantial implications for sea ice as vertical heat transport and storage preconditions sea ice for rapid retreat in coming seasons (Doddridge et al., 2021). Our results also show that Southern Ocean SSTs are shifting, away

from circumpolar cooling in the sea ice zone towards an intensified regional pattern. Compared with earlier decades of the satellite record, the SST anomaly pattern north of the sea ice zone in recent years also alludes to the wave-3 pattern of the asymmetric SAM structure; if both the wave-3 SST pattern and warmer SSTs persist, a rapid sea ice recovery in coming years is unlikely.

### Code and data availability

The NSIDC monthly mean passive microwave satellite sea ice concentration Climate Data Record is available at https://nsidc.org/data/G02202. The daily near-real-time Climate Data Record is available at https://nsidc.org/data/g10016. The

Japanese Meteorological Agency (JMA) global atmospheric reanalysis JRA-55 is available at https://doi.org/10.5065/D60G3H5B. The Met Office Hadley Centre Sea Ice and Sea Surface Temperature dataset version 2 (HadISST.2.2.0.0) is available at https://www.metoffice.gov.uk/hadobs/hadisst/.

## Author contribution

SS designed the study, performed the analysis, created the plots and drafted the manuscript. TO and PS provided important guidance, and all authors discussed and revised the manuscript.

## Competing interests

The authors declare that they have no conflict of interest.

## Acknowledgements

Serena Schroeter was supported by a CSIRO Early Research Career Fellowship. This research/project was undertaken with the assistance of resources and services from the National Computational Infrastructure (NCI), which is supported by the Australian Government.

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

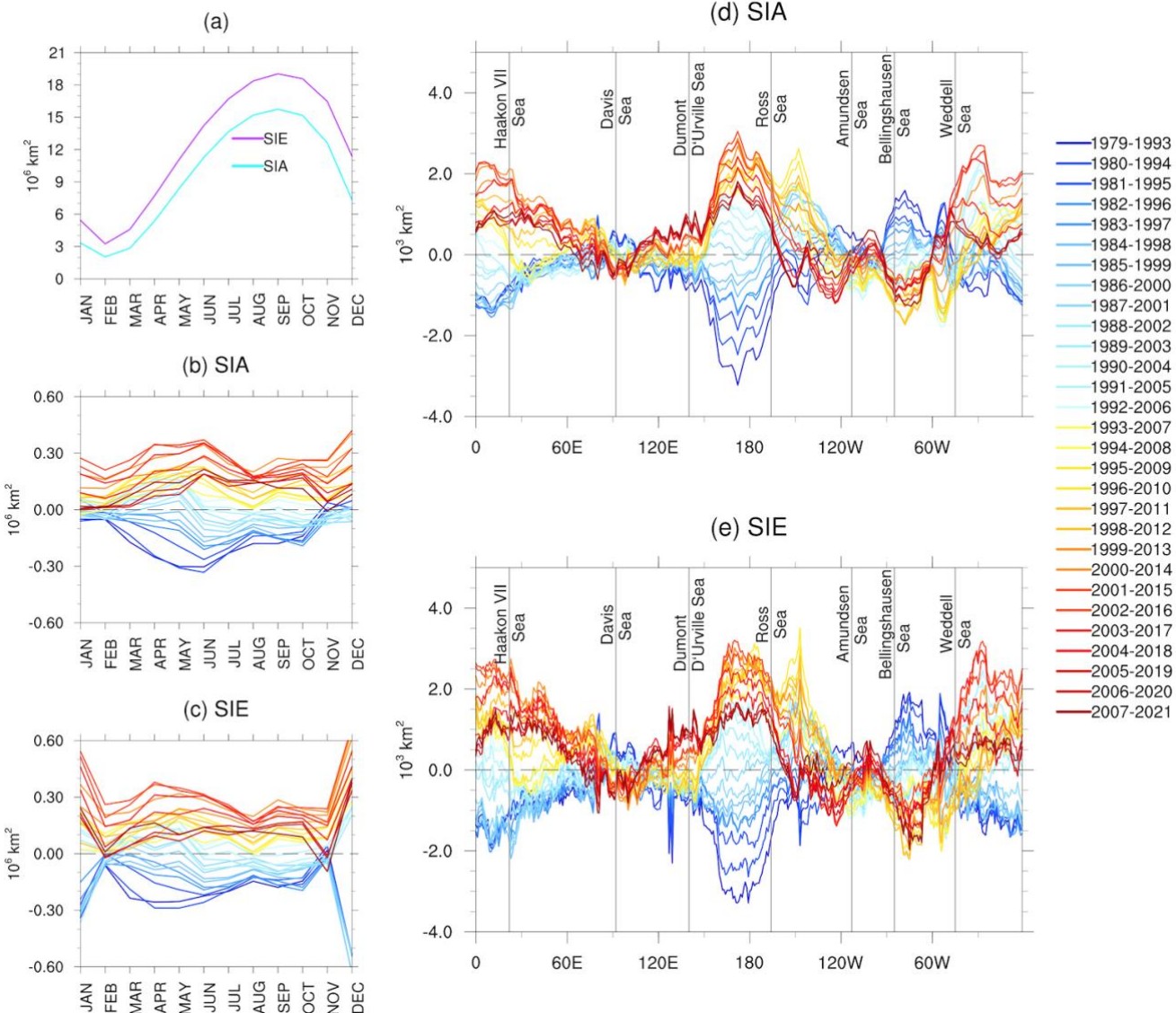

**Figure 1: Running 15-year SIA climatology for (a) integrated circumpolar total SIA and SIE, (b-c) difference of each 15-year climatology from the 1979-2021 climatology of SIA and SIE, respectively, and (d-e) differences of regional 15-yr annual mean SIA**
**and SIE from 1979-2021 annual means. Lines are coloured from earliest (1979-1993, dark blue) to most recent (2007-2021, dark red) 15-yr samples. Vertical lines indicate approximate mid-point of each geographical location; zero difference from long-term mean denoted by horizontal dotted line.**

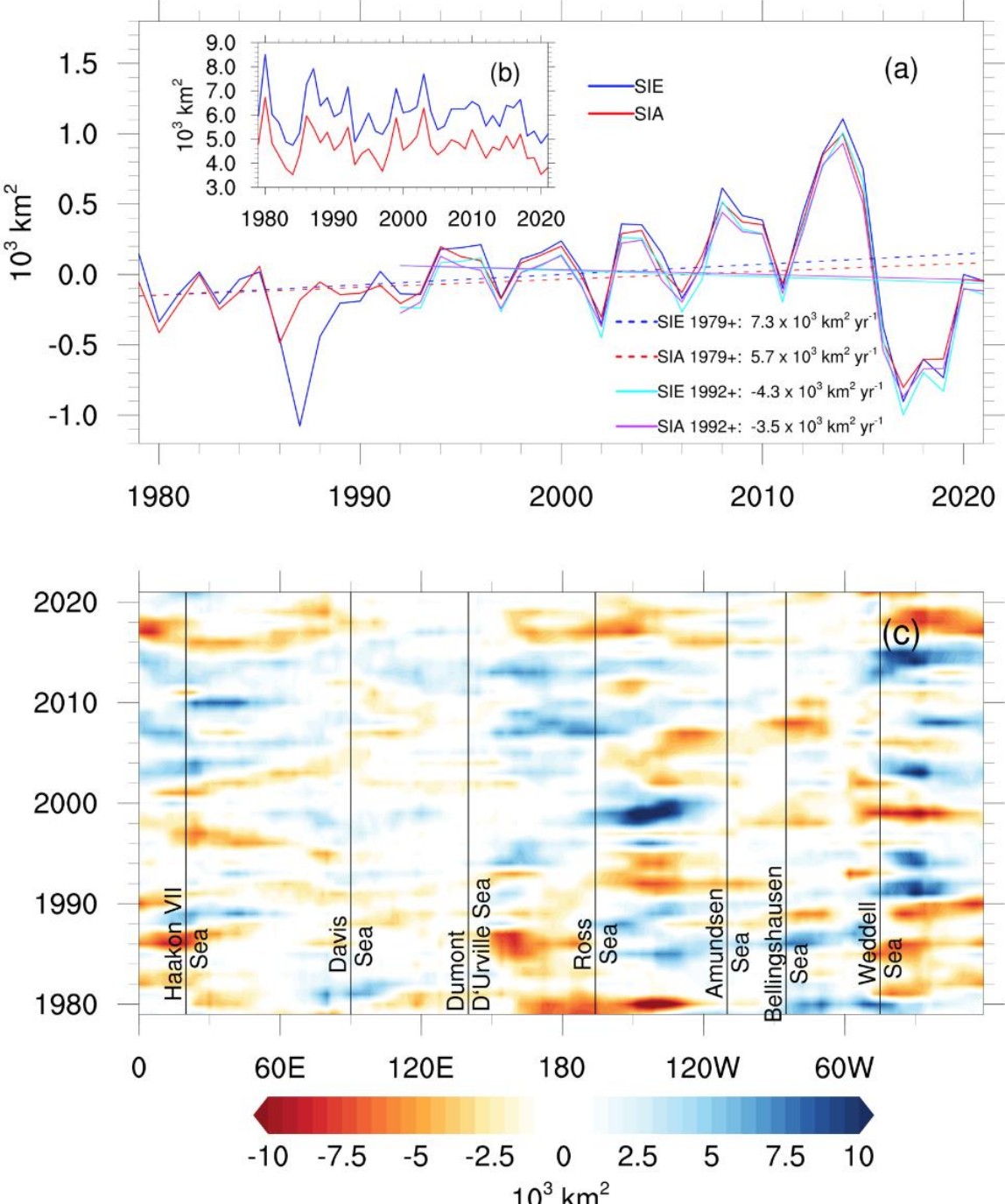

**Figure 2: (a)** Annual average spatially-integrated anomalies of SIA and SIE for long-term satellite record duration (1979-2021) and most recent 30-yr period (1992-2021), **(b)** monthly average standard deviation of spatial anomalies from the 1979-2021 mean, and **(c)** annual average anomalies of SIA from the 1979-2021 climatology. Vertical lines in (c) indicate approximate mid-point of each geographical location.


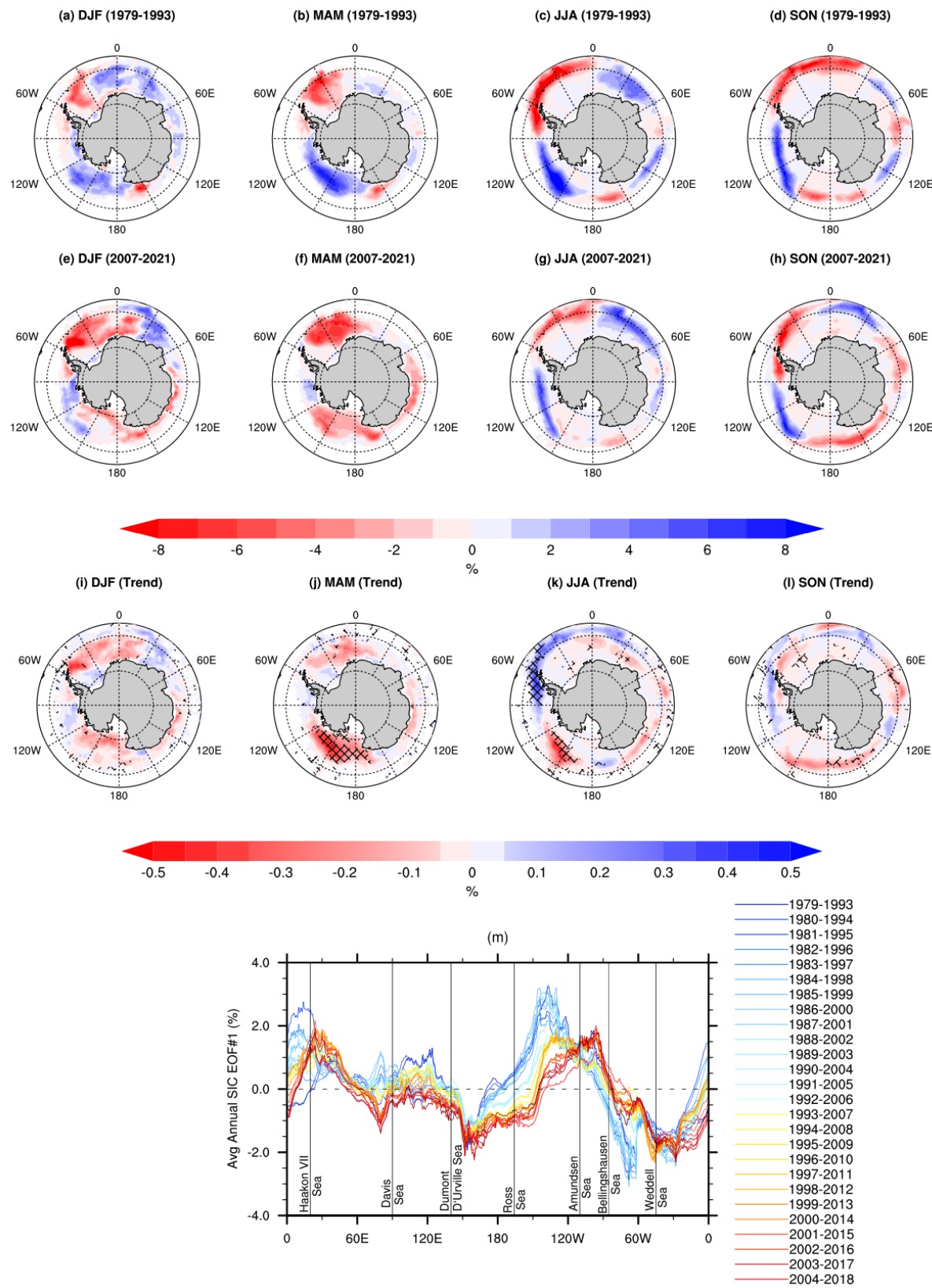

**Figure 3:** EOF1 of detrended seasonal SIC anomalies for (a-d) 1979-1993 and (e-h) 2007-2021; (i-l) trend across running 15-year EOF1 patterns; (m) annual SIC EOF1 averaged over each longitude for each 15-year sample. Lines are coloured from earliest (1979-1993, dark blue) to most recent (2007-2021, dark red) 15-year sample. Vertical grey lines indicate approximate mid-point of each geographical location depicted by vertical text. Stippling shows regions where trend patterns are statistically significant ($p<0.05$).

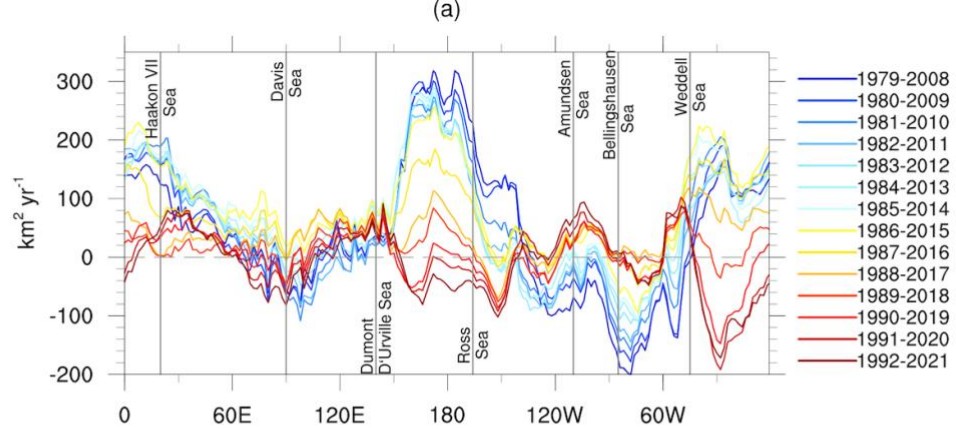

(a)

(b) DJF

(c) MAM

(d) JJA

(e) SON

**Figure 4:** Running 30-year linear least-squares regression coefficients (trends) of: (a) annual SIA anomalies; and (b) seasonal SIA anomalies. Lines in (b) are coloured from earliest (1979-2008, dark blue) to most recent (1992-2021, dark red) 30-year sample. Grey lines indicate approximate mid-point of each geographical location depicted by vertical text. Stippling shows regions where trend patterns are statistically significant (p<0.05).

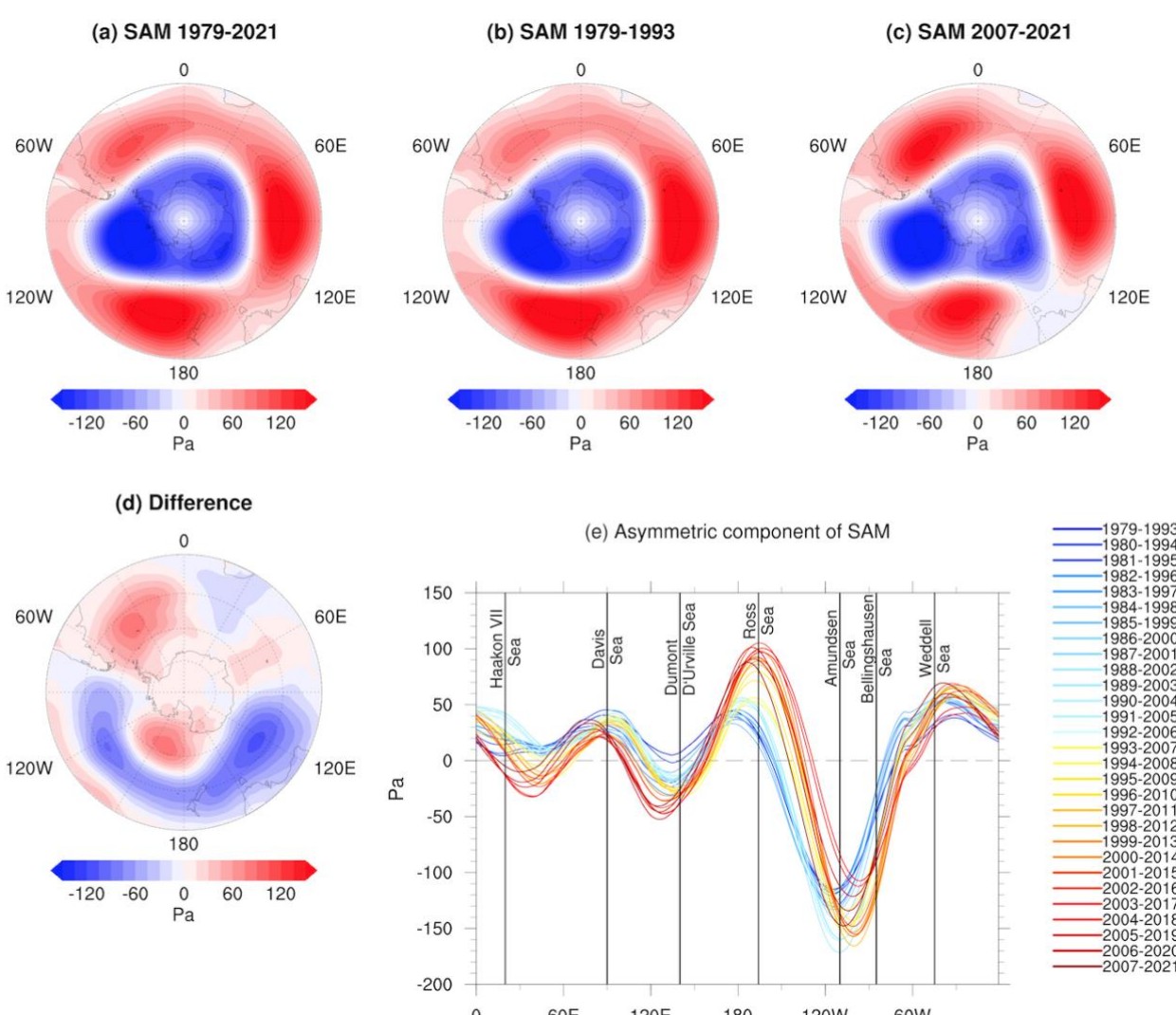

**Figure 5:** EOF1 of detrended SLP anomalies for (a) 1979-2021, (b) 1979-1993, (c) 2007-2021, (d) difference between (b) and (c); and (e) average zonal anomaly of EOF1 between -70°S and -55°S for each 15-year sample. Lines are coloured from earliest (1979-1993, dark blue) to most recent (2007-2021, dark red) 15-year sample. Vertical lines indicate approximate mid-point of each geographical location depicted by vertical text.

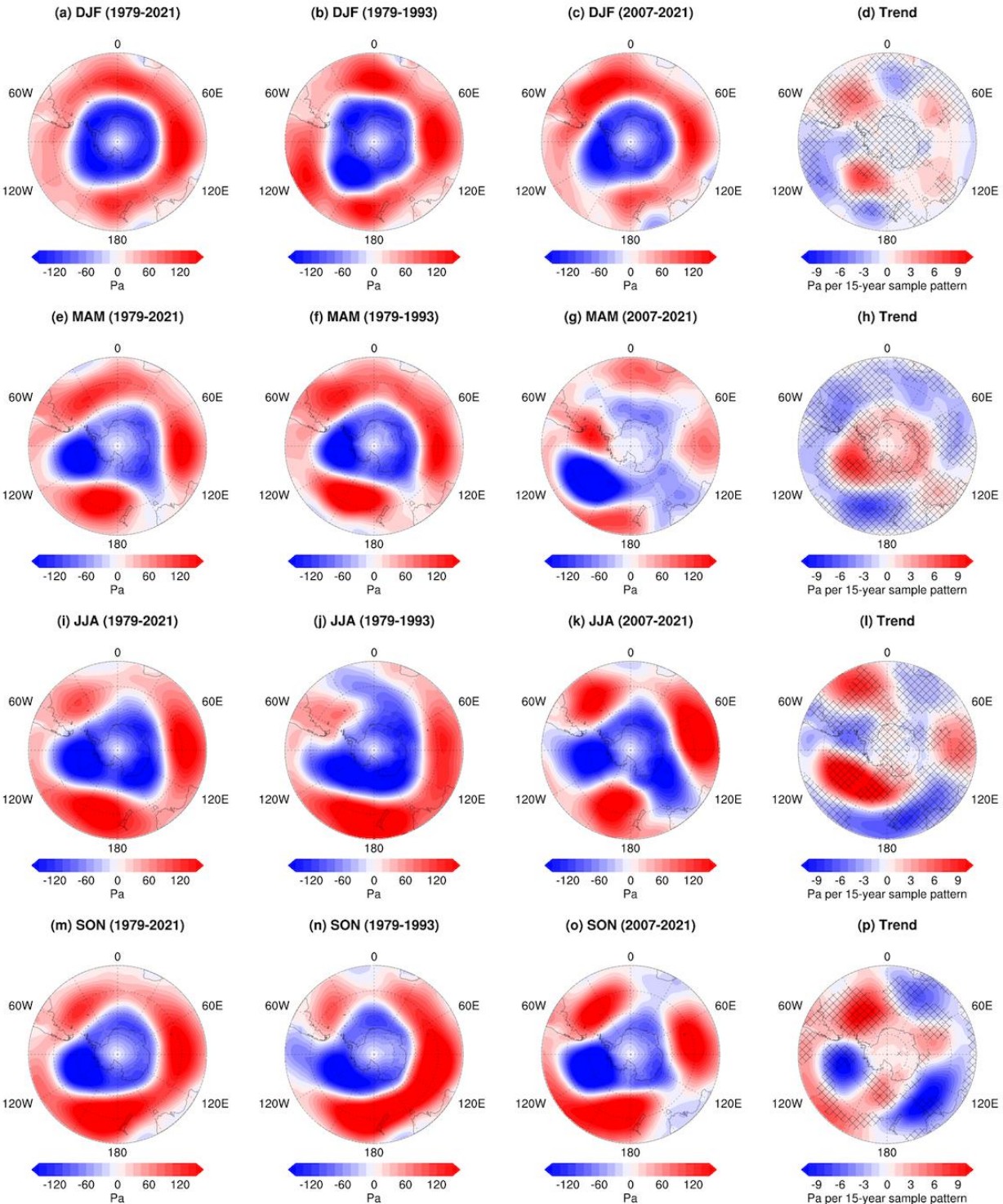

**Figure 6: EOF1 of detrended seasonal SLP anomalies for (a, e, i, m) 1979-2021, (b, f, j, n) 1979-1993, (c, g, k, o) 2007-2021, and (d, h, l, p) trends of running 15-year seasonal EOFs. Stippling shows regions where trend patterns are statistically significant (p<0.01).**

625

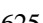

Figure 7: Shading shows the slope of running 30-year seasonal trends of sea ice concentration (SIC) during (a) December-May and (b) June-November, overlaid by contour lines depicting the difference in the asymmetric component of SAM between the earliest 15-year sample (1979-1993) and the most recent (2007-2021). Negative contours (enhanced anticyclonic flow) are shown by dashed lines; positive contours (enhanced cyclonic flow) are shown by solid lines.

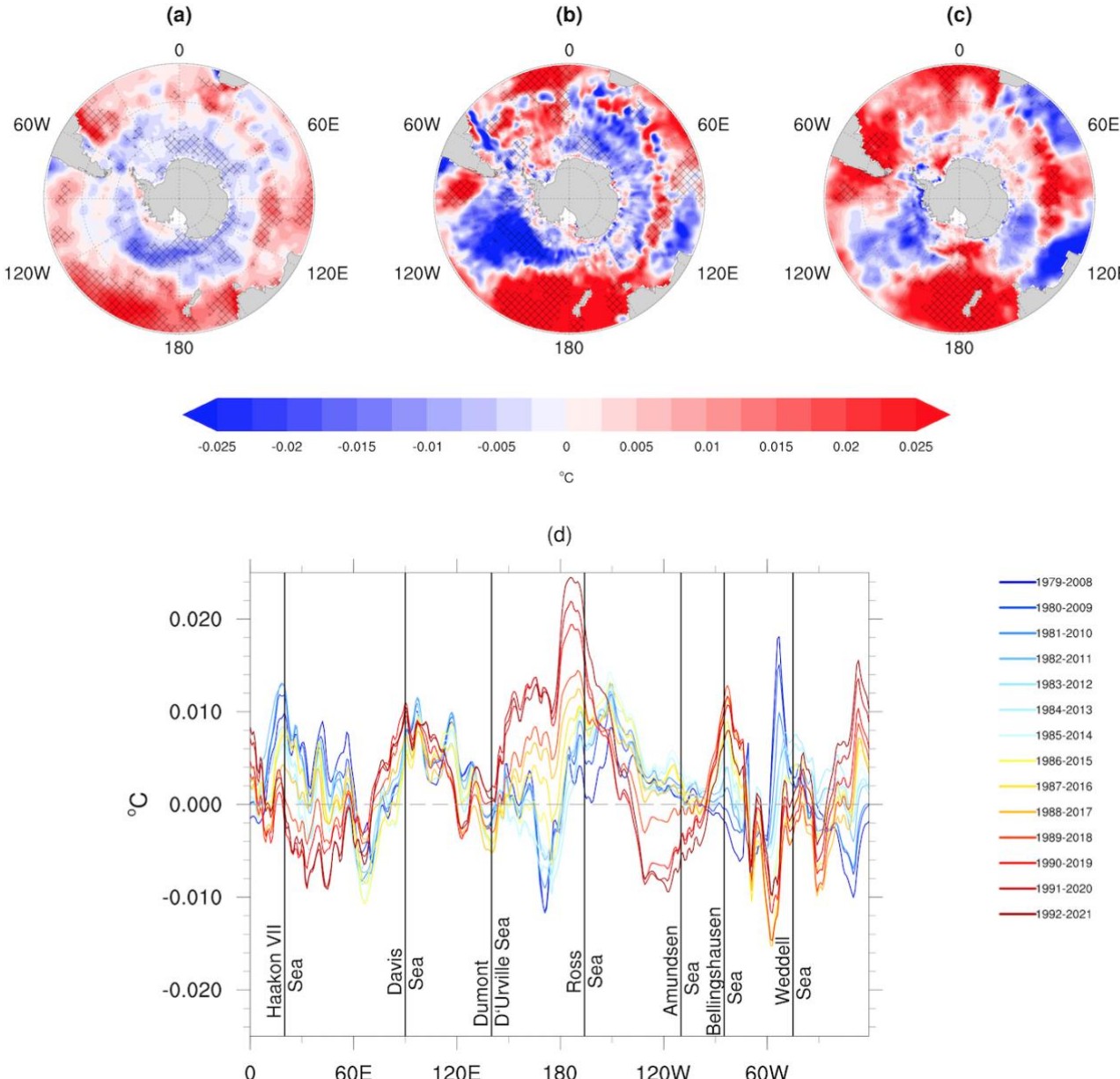

**Figure 8: Annual average regression coefficients of Southern Ocean SST between: (a) 1979-2021; (b) 1992-2021; (c) 2007-2021; and (d) 30-year running samples averaged over the region north of the ice edge (-55°S to -40°S). Lines are coloured from earliest (1979-2008, dark blue) to most recent (1992-2021, dark red) 30-year SST sample. Vertical lines indicate approximate mid-point of each geographical location depicted by vertical text. Stippling shows regions where trend patterns are statistically significant (p<0.05).**