# Peer review of "Antarctic sea ice regime shift associated with decreasing zonal symmetry in the Southern Annular Mode"

_The Cryosphere, 2022_

## Author Response (AR1)

| Reviewer #1 | | |
|---|---|---|
| Reviewer comment | Author reply and/or intended change | Actual change |
| If I understand well, the main point in the discussion is the modification of the link between SAM and the sea ice concentration (Figure 6), associated with the reinforcement in the recent year of the non-zonal component of SAM (Figure 5). What I am missing at this stage is a clear connection with the total changes in ice extent (Figures 1 and 2 for instance). In other words, the manuscript demonstrates a decreased zonal symmetry in SAM, as mentioned in the title. It suggests qualitatively how this impacts the sea ice concentration as the patterns seems to fit with generally an enhanced north-south flow where sea ice extent decrease/increase. However, the fit is not always good and the authors do not quantify how much of the observed changes in the ice extent can actually be attributed to the shift in SAM and this weakens their conclusion. Additional diagnostics or analysis are thus required for me to quantify the proposed links and the impact of the decreasing zonal symmetry of SAM on Antarctic sea ice. | It is true that the fit is not always good – we state within the text that several regions in which SAM asymmetry increases do not show corresponding sea ice anomalies (Haakon VII, Dumont D'Urville etc). In addition, only the coastal, western part of the Weddell Sea shows an agreement with the changes in SAM, due to its position in between the cyclonic/anticyclonic patterns. We are happy to reword this section to more clearly state that the agreement between the patterns of sea ice trends and asymmetry of SAM is only important in regions where SAM is already known to be important – i.e. Ross, Amundsen, Bellingshausen, Weddell seas – and that the other sea ice regions (which have little to no clear signal among the running trends) continue to be dominated by cyclonic activity or other ocean/atmosphere drivers. | We have updated section 3.1 (paragraph 2) to clearly state that sea ice anomalies in the West Antarctic region between the Ross and western Weddell seas, which are known to be sensitive to SAM forcing, agree with the pattern of increasing SAM asymmetry, whereas sea ice anomalies in East Antarctica are dominated by synoptic weather and therefore show no clear, corresponding response to increasing SAM asymmetry. |
| | We (the authors) understand that perhaps the first part of the results section and the final discussion as well need to make clearer that it is not our intention to detect and attribute regional changes as a proportion of total sea ice change. In showing Figure 1 and 2, our main intention is to highlight the importance of a regional perspective in understanding how Antarctic sea ice variability and trends have changed over time, as it is very common for studies to use circumpolar total metrics or sector averages which may mask that trends or patterns of variability can (as seen in Figure 3) shift spatially as well as temporally. | The text has been substantially revised with seasonal and regional perspectives, statistical significance, and updated/new figures to more clearly link changes in SAM and Antarctic sea ice. |
| The origin of the shift in SAM is not discussed. I understand that it is not the subject of the paper. However, the reader would be interested to know if this could be due to multi-decadal variability in the atmospheric circulation, a response to the greenhouse gas forcing, recovery from the ozone hole, or any other mechanism. I would thus suggest to add a paragraph in the final section, at least to present the different hypotheses. | Agreed – we are more than happy to include a paragraph or two as needed to more thoroughly discuss the shift in SAM and its potential drivers, and agree that this would add much-needed context for the discussion. | The paragraph in the discussion section discussing the shift from blocking pattern to zonal SAM has been augmented to include a discussion of the drivers of the shift towards positive SAM polarity and the potential implications of ozone recovery. |
| For several diagnostics, I was not totally sure of the diagnostics that is displayed. As this is key for understanding the paper, I would recommend to add more details on the way they are produced. In particular, for figure 2b, that would be useful to explain exactly how regional variability is computed. For figure 4 and 7, regressions of SIA anomalies and SST are mentioned but I am not sure with what SIA or SST is regressed, maybe with time? Can 'regression' here be considered equivalent to 'trend'? If this is the case, I think that it would be useful to specify it. Figure 6 mentions a gradient of 15yr samples of average EOF1 zonal anomaly. I guess it is the zonal gradient and thus the derivative of the plot of Fig. 5d but I am not totally sure. | Thank you for pointing this out. We agree, and would expand upon the methodology section, figure captions, and within the text as needed to clarify the calculation of each diagnostic and its physical meaning. | The method section has been updated to more clearly state what is being calculated and displayed, and in-text discussions and figure captions include greater detail as well. |
| In many places, starting in the abstract, the authors mentioned that the long-term trend is skewed towards the earliest years (line 9; line 83, line 211). Maybe 'skewed' can give the feeling that the estimate of the long-term trend is wrong or biased. I would personally preferred 'dominated by the changes in the earliest year' or something equivalent that is more neutral. | No problem, we can rework these sections to remove any potential negative connotations of wording choice. | Wording changed throughout to remove "skewed". |
| Line 113. The eastward shift of the anomalies is interesting for me. The authors interpret the changes in patterns as an increase in the meridional exchanges related to SAM but would it be possible that a part of the signal is due to and eastward shift of the pattern or to the advection of some anomalies by oceanic currents. On this subject, maybe a link with the very recent study of Morioka et al. (2022) would be interesting. | Thank you for the reference, we will closely examine the aforementioned study and incorporate it into the text both in the section listed and also in the final discussion. It is entirely possible that the signal in this region is partly due to oceanic current anomalies – we do not claim SAM is responsible for the entirety of anomalous sea ice in any region. Indeed, the Morioka et al. (2022) study does state in the conclusions that the weak 6-10yr prediction skills in the Ross Sea could be (at least in part) due to the influence of the SAM in this region on decadal timescales, so fits well with our discussion of what drivers may be shifting these patterns of variability. | The Morioka et al. 2022 study is included in a section further down, discussing the potential drivers of the eastward shift of anomalies as well as the fact that no one single process is likely responsible for driving sea ice trends. |

| Reviewer comment | Author reply and/or intended change | Actual change |
|---|---|---|
| Line 244. It is indeed counterintuitive and I do not follow well the argument here. Line 243, it is said that increasing meridional flow over the sea ice zone is driving spatially heterogeneous anomalies. If I understand well the sentence, a higher overall variance of the total sea ice extent would be due to a 'greater agreement across regions of high-magnitude changes'. If this is the case, that would be important to quantify this. | We agree that this is not clearly worded and that this paragraph requires rewording. Figure 2b and c show that, compared with the widely-opposing anomalies in early years of the satellite record, zonal sea ice variability since about the mid-2000s is much more moderate, and variability further drops in the most recent years because the anomalies in previously strongly-opposing regions now mostly agree. While this might indicate a driver that produces a spatially heterogeneous sea ice response, the variability in recent samples is low but still not zero. The trend magnitudes in the most recent samples are quite small compared to the early samples, and the only statistically significant regional trends in the 1991-2020 and 1992-2021 samples are small increasing trends in the Amundsen and Dumont D'Urville seas, not any of the larger decreasing trends in the Ross or Weddell seas. Spatial heterogeneity is still therefore present, but the magnitudes are low in the context of the long-term shift in these trends, and the spatial pattern is almost the opposite of the earliest (1979-2008) pattern. Since the SAM change has not been a sudden change, but rather a gradual shift, the sea ice response would also be gradual – especially since there may be multiple factors affecting sea ice variability on shorter timescales, as you've pointed out above. We (the authors) intend to make it much clearer in the discussion text that the changes in SAM only strongly affect the regions where SAM variability dominates sea ice | This section has been replaced with a paragraph explaining the cancellation effect of seasonal anomalies and trends, and how the patterns do in fact reflect the changing SAM pattern when examined spatially and seasonally. |
| Line 247. Why is it assumed that the changes that are underway are likely to continue? The response of sea ice to wind changes is usually relatively fast. If it is because it is assumed that the asymmetric flow pattern will continue to intensify, this should be explained in more details here. | Agreed. It is based on whether the asymmetric flow pattern continues or intensifies, and an enhanced discussion of SAM trends, potential drivers and variability (as you've already suggested) should aid this substantially. The response of sea ice to wind changes is indeed quite fast, but as mentioned above, the SAM change is not sudden. We (the authors) will also make it clearer that SAM is not the only – or even most important – driver of change, especially outside of the Ross/Amundsen/Bellingshausen/Weddell, so especially around East Antarctica these wind changes may not provoke much sea ice response at all. | Sentenced removed; SAM trends with greenhouse gas etc discussed in a different section. |
| Figure 5. The caption does not seem to correspond to the figures. The period 1979-2001 is mentioned in the figures (panel a) but not in the caption. Is the difference (panel d not c) between (a) and (b) or between (b) and (c) ? | Apologies – somehow this outdated caption was missed during proof-reading. The caption will be corrected. Thank you for bringing this to our attention. | Caption updated. |
| | Reviewer #2 | |
| Reviewer comment | Author reply and/or intended change | Actual change |
| More detail is needed throughout – particularly if the analysis is done on annual means or monthly anomalies, or something else (Fig 3, Fig 5-7). More details are needed on the calculation of the SAM gradient, as well as some demonstration of the relationship with the meridional wind and its changes through time. Important seasonal deviations from the annual mean or annual cycle (the latter for sea ice especially) are needed throughout. (some details are provided in the listed minor comments below) | Thank you for pointing this out. We agree that inclusion of measures of statistical significance where possible improves the discussion. Our intention is to incorporate a method of calculating linear least-squares regression coefficients in order to account for temporal autocorrelation, by modifying the calculations of residual variance and standard error to use an effective sample size based on lag-1 autocorrelation coefficients (Santer et al. 2000). A couple of sentences on the methodology would of course be included in the methods section. Statistical significance would be calculated for all regression coefficients in all figures, and contour plots also included so as to overlay statistically significant regions with stippling to aid interpretation. The text and captions would also be updated accordingly. We believe this strengthens our analysis: initial attempts indicate that, for example, the statistically significant regionally-opposing trends of SIA occur early in the satellite record, and in 30-year samples from 1988 onwards, the Ross Sea trend becomes statistically insignificant and weakly reverses sign while the Amundsen Sea trend reverses sign and becomes statistically significant in the most recent samples. | The text has been substantially reworked with the updating of methodology, more detail of calculations and conclusions within the text, and augmentation of figures with additional seasonal and regional perspectives. The lineplot with the SAM gradient has been changed to instead overlay contours of the asymmetric SAM change on top of contours of the slope of SIC trends to more clearly show the relationship. |

| | | |
|---|---|---|
| While the shorter time period helps to understand changes in anomalies or trends, there is essentially nothing done in the manuscript to discuss any statistical significance. I suspect as the time period / sample size decreases, the relationships are not statistically significant. In my view, this needs to be discussed – and changes that are statistically significant need to be emphasized. While I realize the paper is about the large variability – it needs to be made clear that this large variability dampens the ability to detect meaningful relationships beyond noise, especially at smaller temporal scales (smaller sample sizes). | Greater detail will also be included for the methodology of diagnostics and analysis, particularly the SAM gradient. We accept that this is unclear at times in the text. We are also happy to include seasonal perspectives, whether in place of or alongside annual perspectives where necessary, or as supplementary material. | Statistical significance calculated for all regression coefficients, with temporal autocorrelation accounted for as discussed; method section updated to reflect this. Figure 4 augmented to include a contour plot of 30-year running regression coefficients (trends) with stippling overlaid to show regions/temporal samples that are statistically signfiicant. |
| L25: A recent paper by Turner et al. (2022) may be worth citing here as well | Thank you for supplying this reference, this will be included. | Reference included. |
| L69. 72: change 'data is' to 'data are' | The wording will be changed as suggested. | Wording changed. |
| Fig 1a – I think you have the SIA and SIE lines labeled incorrectly, SIA should be larger than SIE | Actually, by standard convention, SIE is always larger than SIA, because SIE counts the total grid cell area of any cells containing at least 15% concentration, whereas SIA only counts the ice-covered fraction of any grid cell (see Notz, D., 2014. Sea-ice extent and its trend provide limited metrics of model performance. The Cryosphere 8, 229-243, doi:10.5194/tc-8-229-2014, introduction: paragraph 4). | None. |
| Fig1b-e, and discussion on lines 79-83: Perhaps it is better to call these 'anomalies' rather than deviations, since from the word deviation I immediately think of standard deviation (a measure of variance – which is clearly not what you are showing), rather than a difference from the mean | No problem, we'll change the wording throughout the paper to be "differences" from the mean instead to avoid confusion, only using "deviation" in relation to standard deviation as suggested. | Wording changed. |
| L83: again, from statistical standpoint, suggest changing to: '…the long-term mean of both SIA and SIE is more reflective of conditions during the earliest years of the satellite observations' as you are not showing the statistical measure of skew | No problem, we'll change the wording to be "dominated by" rather than "skewed towards" so as to avoid negative connotations. | Wording changed. |
| Fig. 2 – again, SIA should be greater than SIE (SIE does not include ice with less than 15% concentration), so something is off in the labeling here | See response above; SIE is always larger than SIA. | None. |
| Fig 2b- It doesn't appear that the trends in sea ice variability are statistically significant, and indeed, if there is any decrease it is because the higher variability at the end of the timeseries – much of the timeseries shows relatively consistent (at least for such a highly-variant sea ice!) year to year variation. | Agreed. We're not arguing that there is a statistically significant trend in variability (which we will explicitly stated in the text for clarity). Figure 2b is intended as both an illustration of the generally high variability and, as you say, that it is high at one end and low at the other. This contrasts with the annual average anomalies of Figure 2a, in which the anomalies are low at one end and high at the other, because of the cancellation effect of the high variability in early years and the agreement between key regions in more recent years, aggregating the spatial domain of Figure 2c. | This paragraph has been reworded to include a more careful discussion of the variability plot, and the trend lines removed from 2b. Instead, the text points out the 'peaks' and 'troughs' of the earlier half of the satellite record, compared to the relatively moderate level of variability in later years, and the low values between 2018-2021. |
| Fig 3 – are these EOFs based on all monthly anomalies, or for the annual mean? Either way, it masks the important seasonal cycle in sea ice and therefore makes it difficult to interpret. In the same way your paper argues that the climatological (time) mean is biased toward the earlier part, these EOFs represent conditions of sea ice that are only observed for a small portion of the annual cycle. They don't appear to be a robust representation of the dataset. It would be better to look at conditions perhaps at sea ice max / min, or seasonal averages instead. | Thank you for this comment. The EOFs are calculated from monthly anomalies. The methodology section will be detailed further to clarify the calculation of EOFs. We agree that the analysis would benefit from a seasonal perspective here, and would include seasonal average EOFs (perhaps surrounding the minimum and maximum as suggested, or the most robust representation as possible based on monthly EOFs). | Figure 3 has been modified to show 4 seasonal EOF patterns for the first and most recent 15-year samples, as well as the linear least-squares regression coefficients (trend) across the samples, with adjusted statistical significance overlaid. The line plot of the previous Figure is retained for visual representation of the consistency of the shift in some regions. The method has been updated to more clearly outline how the EOFs are calculated, and some extra detail has been included in the section itself for ease of reading. |
| Fig 3 – the correlation has shifted perhaps, but again due to the very small sample size there is not a statistically significant shift in the correlation magnitude. | Is this in reference to Figure 3c? The pattern correlation results of each 15-year EOF against the most and least recent EOFs is intended to show the gradual change over time; however, if this is not useful, we can remove this panel in favour of a more detailed seasonal perspective as suggested in the previous comment. | Panel removed in favour of seasonal analysis described in previous point. |

| | | |
|---|---|---|
| Fig 4 – the challenge in interpreting the changes in these long term trends (again masking the annual cycle) is that a lot of the sea ice is gone in the Bellingshausen and King Hakon seas in recent years – which naturally would decrease the variance and weaken the spatial heterogeneity. I'm not entirely convinced from these figures that it is a sea ice regime shift rather than just a complete (or near complete) removal of much of the summer sea ice in these regions and a lengthened ice-free season. The pattern seems to be preserved in the Ross Sea and Weddell Seas, areas with the most ice (even in recent years) in the austral summer seasons. | Thank you for this comment. We agree that inclusion of seasonal trend patterns here as contour plots with statistical significance stippling overlaid would aid interpretation and add important seasonally compensating information to the discussion. An initial test of this clearly shows gradual versus abrupt shifts in the rolling trend patterns, and also shows a somewhat wave-3-like pattern in trend contours in recent samples during JJA and SON. The removal of summer ice/lengthened ice-free season is an important point, and one we would explore further by examining seasonal or monthly anomalies and trends. | Figure 4 has been modified to show 4 seasonal trend plots as contours with statistical significance stipling overlaid, as well as the original annual average trend lineplot. The seasonal contours clearly show an eastward shift of the positive trend pattern from the Ross towards the Amundsen/Bellingshausen during winter and spring, as well as the emergence of a wave-3-like pattern in sea ice trends in the most recent patterns during the same seasons, when SAM has the strongest influence on sea ice. The contours also show the stopping of positive trends in the Ross Sea and the trend sign reversing in the Weddell seas in recent 30-year samples during the austral summer and autumn (except for the western boundary of the Weddell Sea), along with a (statistically significant for the last two samples) positive trend in the Amundsen/western Bellingshausen in DJF spanning the last 6 samples. |
| Fig 5 – is this for annual means? If so, you are masking the role of ozone depletion that has a strong seasonal footprint on the SLP anomalies (which aren't really reliable over Antarctica, but I suspect you'd have a similar depiction in surface pressure anomalies) – this should be noted | No, we do not calculate the EOFs from annual means, the SLP EOFs are calculated on detrended, cosine-weighted monthly anomalies from 15-year running monthly means. This will be given a more detailed description in the method section for clarity. | Figure 5 uses monthly anomalies, and an additional Figure has been included that uses seasonal anomalies to show the different spatial patterns and tendencies across the short-term samples. The methodology for calculating the EOFs has also been expanded for clarity. |
| Fig 5 – the caption needs redone as it does not describe the paneling in the actual figure – there's no 5e in the caption (it is incorrectly described as 5d) | Apologies – somehow this outdated caption was missed during proof-reading. The caption will be corrected. Thank you for bringing this to our attention. | Caption updated. |
| L157 – the SAM structure may be more zonally asymmetric in the annual mean, but it is more symmetric in the summer according to Fogt et al. (2012) – likely due to the role of ozone depletion mentioned above | We agree that this section would benefit from a seasonal perspective on changes to SAM symmetry, and would seek to add this to the existing figures and discussion. | An additional Figure has been added depicting the long-term (1979-2021), earliest (1979-1993) and most recent (2007-2021) short-term EOFs for each season (DJF, MAM, JJA and SON), as well as a plot showing the trend of the pattern across the 29 15-year sample patterns, with stipling overlaid to show statistical significance of the trend pattern (p < 0.01). Section 3.1 has been augmented to include a discussion of this additional Figure and the seasonal changes in SAM on short-term timescales. |
| Fig 5e – can you confirm that the zonal anomalies sum up to zero for each 15 year period? The integration almost looks negative for the later part (red lines), but it could be my eyes tricking me. | Confirmed – every sample sums to zero across the longitudes. | None. |
| Fig 6 – how is this gradient calculated exactly and it is smoothed in some way spatially? | We agree that this needs to be more clearly stated in the methods section. EOFs are calculated from 15-year samples of detrended anomalies that have been weighted by the cosine of latitude (to compensate for meridional convergence near the poles). In all samples, the symmetric SAM component is then calculated as the zonal mean of EOF1, while the asymmetric SAM component is calculated as the zonal anomalies which are then averaged over the latitudes -70°S:-55°S (encompassing much of the sea ice zone through to 55°S as the region of maximum amplitude for ZW3). The gradient is then calculated simply as the difference between the most and least recent patterns, divided by the number of samples. We will consider whether this is the most appropriate metric to show this; a regression coefficient of the 15-year ASAM samples shows a similar pattern but offset (for example with a trough in the eastern Ross rather than slightly to the west, and a peak over the Bellingshausen instead of the Amundsen), which more clearly captures the edges of the intensified wave pattern. | The methodology for calculating the EOFs has been expanded and clarified, and the asymmetric SAM component has been detailed in the body of the text of Section 3.1. |
| Fig 6, lines 170-176: Is there are way to show these relationships differently, and to demonstrate some of level of statistical significance? Correlations of SIE anomalies with meridional winds or something similar? You talk about things being in agreement (with meridional winds in particular), but none of this is directly shown other than the difficult to interpret connections with the SAM gradient. | Thank you for this point. We will consider whether meridional wind trends can be used as well in order to show statistical significance, and as stated above will also consider whether the SAM gradient alone is an appropriate metric to show this change. | Instead of the gradient line of asymmetric SAM overlaid on the SIA trend lineplot, we show the slope of running 30-year seasonal SIC trends as polar stereographics with contours of asymmetric SAM overlaid, which is a much clearer picture of the changes and locations of the regions of north-south flow. |
| L220-229: Worth adding into the discussion here the recent paper by Fogt et al (2022) who also discuss a regime shift in Antarctic sea ice in the 20th century – consistent with the lower SIE in the mid 20th century near the start of the satellite observations (https://www.nature.com/articles/s41558-021-01254-9) | Thank you for supplying this reference, we will include this as suggested. | The paragraph has been augmented to include the Fogt et al. 2022 study. |